# A Co-Adaptation Method for Resilience Rebound in Unmanned Aerial Vehicle Swarms in Surveillance Missions

Kunlun Wei [iD], Tao Zhang *[iD] and Chuanfu Zhang *[iD]

School of Systems Science and Engineering, Sun Yat-sen University, No. 135, Xingang Xi Road, Guangzhou 510275, China; weiklun@mail2.sysu.edu.cn
* Correspondence: zhangt358@mail.sysu.edu.cn (T.Z.); zhangchf9@mail.sysu.edu.cn (C.Z.)

**Abstract:** An unmanned aerial vehicle (UAV) swarm is a fast-moving system where self-adaption is necessary when conducting a mission. The major causative factors of mission failures are inevitable disruptive events and uncertain threats. Given the unexpected disturbances of events and threats, it is important to study how a UAV swarm responds and enable the swarm to enhance resilience and alleviate negative influences. Cooperative adaptation must be established between the swarm's structure and dynamics, such as communication links and UAV states. Thus, based on previous structural adaptation and dynamic adaptation models, we provide a co-adaptation model for UAV swarms that combines a swarm's structural characteristics with its dynamic characteristics. The improved model can deal with malicious events and contribute to a rebound in the swarm's performance. Based on the proposed co-adaptation model, an improved resilience metric revealing the discrepancy between the minimum performance and the standard performance is proposed. The results from our simulation experiments show that the surveillance performance of a UAV swarm bounces back to its initial state after disruptions happen in co-adaptation cases. This metric demonstrates that our model can contribute towards the swarm's overall systemic resiliency by withstanding and resisting unpredictable threats and disruptions. The model and metric proposed in this article can help identify best practices in improving swarm resilience.

**Keywords:** UAV swarm; system resilience; metric; co-adaptation; communication links

## 1. Introduction

A UAV swarm comprises many UAVs equipped with specific autonomous capabilities that enable them to successfully carry out assigned missions through real-time information exchange and collaborative cooperation. A typical application scenario for a UAV swarm is surveillance coverage, which includes tasks such as forest fire monitoring, search-and-rescue operations, and military reconnaissance [1]. However, during surveillance tasks, some UAVs may encounter disruptions coming from harsh environments or deliberate adversaries. These disruptions can lead to a decrease in surveillance coverage and hinder the overall completion of the assigned tasks. Highly reliable components, redundant designs, and other traditional methods have proven ineffective in effectively and promptly dealing with unpredictable threats and undesirable events [2]. However, it is anticipated that UAV swarms could be able to withstand disruptive events, handle accidents, and restore their intended performance through self-adaptation [3].

In a UAV swarm, resilience refers to the system's ability to keep a desired performance despite incidents that may disrupt its normal operation [4]. Resilience offers a new perspective on enhancing and improving systems' ability to bear unforeseen events and recover from disruptive threats. Therefore, resilience is a suitable evaluating concept for a UAV swarm in situations where incidents inevitably occur. The evaluation of the resilience of a swarm primarily focuses on two key aspects: developing an improved network model that accurately describes disruption and recovery events and proposing an appropriate metric for evaluating resilience.

A well-designed topological structure for UAV swarms can greatly enhance their surveillance capabilities [5]. However, most resilience studies published so far have only focused on the recovery ability of UAV network structures for inevitable threats and disruptive events [6,7]. Unfortunately, these studies have limited capability to determine whether the swarm can effectively carry out its assigned surveillance task. To address this gap, Dui et al. [8] demonstrated that the reliability of UAVs in different locations has varying effects on swarm performance, and they discussed topology enhancements for UAV swarms based on importance metrics. Additionally, Xu [9] identified node movement as a key factor impacting network performance and developed a novel model for information exchange networks in unmanned swarm systems. Furthermore, Bai et al. [10] investigated the topology characteristics of UAV swarms, taking into consideration the influence of limited communication range, and proposed an improved UAV swarm model. They also presented an effective method to mitigate the adverse effects of unforeseen threats and disruptive events through topology self-adaptation in UAV swarms. Those studies attempted to utilize the structure self-adaptation method in UAV swarms, but self-adaptation was found to inevitably cause structure and state dynamics problems.

Some studies have focused on the dynamic nature of swarms. For instance, Liu et al. [11] examined multistate systems, wherein their network and component status follow their probability distributions. Hu et al. [12] introduced an innovative form of UAV network based on blockchain and a software-defined network architecture. This design enabled the rapid construction of swarm networks while meeting application requirements and communication circumstances. In [13], a centralized training and decentralized enforcement mechanism was implemented. This approach utilized data gathered by all UAVs to train shared control programs. Each UAV made decisions according to the observational information it collected. Additionally, Wang et al. [14] proposed a dynamic and distributed method for UAV control based on formation, specifically targeting the connectivity and coverage of spatially dispersed users. They utilized flocking dynamics to develop detailed plans for UAVs during mission execution. These studies have primarily focused on the dynamic characteristics of swarms, which can partially but not comprehensively meet resiliency requirements [15].

Recent studies have highlighted the significance of interdependent interactions between dynamic characteristics and network topology in UAV swarms [16]. There is a growing trend towards incorporating resilience considerations into the study of UAVs, particularly in relation to topology and dynamics. George et al. have put forward a groundbreaking framework that adopts a comprehensive approach to resilience design. This framework enables a system to effectively monitor both external and internal disturbances, detect and anticipate threats, and take appropriate actions to counteract any performance-related issues [15]. In a similar vein, Hu et al. decoupled the data plane from the control plane of each UAV and proposed a practical approach [12]. Additionally, Mou et al. developed a trajectory planning algorithm based on graph convolutional neural networks that enables the rapid reconstruction of communication links during the self-healing process [17]. A UAV swarm's structure and dynamics are two sides of the same coin; as such, it was found that structural self-adaptation and dynamic self-adaptation could not meet the requirements for swarm resilience independently. Considering the cooperation of structure adaptation and dynamic adaptation gives us a feasible direction.

The resilience metric has recently been proposed and applied to multi-UAV system-of-systems (SoS) [18,19]. Tran defined the total received information in an information exchange network at any given time as the system performance [2]. Based on Tran's work, Bai et al. [10] proposed an improved resilience metric, where the standard performance of the swarm was calculated using different methods. The operation loop was described as the working process of an unmanned weapon, representing a specific instance of the primary operation loop. Sun et al. [20] used the quantity of operation loops as a comprehensive metric to characterize the performance of unmanned weapon SoS. Based on information traffic load balancing, Zhang et al. [21] proposed a resiliency evaluation framework for

UAV swarms as well. Zhang et al. [22] utilized complex network theory to analyze network topology, using topological metrics such as the clustering coefficient, average node degree, and network efficiency as indicators of system performance. Li expanded on the generalized baseline function of information exchange capacity in communication networks, and also considered the heterogeneous communication network topology of UAV swarms. Compared with Tran [2] and Bai [10,23] incorporated functional factors into the exchange capacity model for network information. The majority of studies on surveillance missions predominantly concentrate on achieving optimal coverage area [24,25], the recognition and tracking of multiple targets [26], and the effective distribution of UAVs in a surveillance area [27].

The above resilience and performance metric may not be suitable for the UAV swarm performing surveillance task. In most cases, UAV swarms are required to carry out a mission successfully, so the minimum performance should be capable of accomplishing the designated surveillance mission [28,29]. The commander or administrator may wish to grasp the differences between the real-time performance and the standard performance of a UAV swarm after a disruption occurs. It should be noted that the total information quantity metric of a swarm system received at time $t$, as a metric of swarm performance, is unable to provide insights into the characteristics of a surveillance mission [30]. A significant portion of the information transmitted in the swarm is irrelevant to conducting the surveillance mission. The information metric cannot reflect the situation of overlapping surveillance areas, which is common in surveillance missions [31,32]. The validity of using the shortest path method [9,33] to calculate the total amount of information in the swarm as a performance indicator requires further proof in practical scenarios [34].

It is imperative to propose a suitable performance and resilience metric to accurately measure and evaluate a UAV swarm's ability to perform a surveillance mission [35]. In order to compensate for the shortcomings in the performance and resilience metric [9,33] of the surveillance mission, we utilize the surveillance area of each individual UAV at time $t$ as a measure of its performance, and the total surveillance area is used as an indicator of the swarm's overall performance [32,36]. By employing this metric, we aim to accurately assess the capacity of a UAV swarm to carry out surveillance missions. The main contributions of our research are summarized as follows:

I. We improve the adaptation models of ecosystems and propose a static model, structural adaptation model, dynamic adaptation model, and co-adaptation mode for UAV swarms to resist disruptive events and uncertain threats and ensure performance recovery.

II. We propose a new performance metric for UAV swarms executing an surveillance mission. Based on the surveillance mission, we also provide an algorithm to compute this performance metric.

III. We develop an improved resilience metric for UAV swarms based on the reported resilience metric.

The remainder of this paper is organized as follows. Section 2 presents four different models of a UAV swarm. In Section 3, we investigate an improved resilience metric. Section 4 verifies the proposed model through several illustrative experiments. In Section 5, we give our conclusions and future work.

## 2. Adaptable Model of the UAV Swarm

Figure 1 depicts a sketch map of a UAV swarm, each UAV is characterized as a node and distinguished by numbers. The nodes size denotes the surveillance scope of UAVs and the black lines are communication links between UAVs.

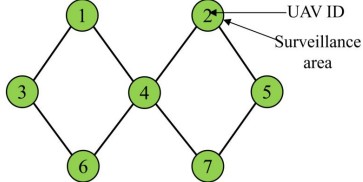

**Figure 1.** A sketch map of a UAV swarm.

The interaction weight of UAVs ($i$, $j$) is determined by the number of $j$s connected neighboring UAVs. The weight between ($i$, $j$) becomes weaker as the neighboring UAVs number of $j$ increases. UAV $j$ contributes $M_{ij} / \sum_{k=1}^{N} M_{kj}$ to $i$, where the denominator is the total number of $j$'s neighbors. $M_{ij}$ represents the number of mutually connected relationships between $i$ and $j$. For each pair ($i$, $j$), the contribution of the interaction weight $A_{ij}$ between $i$ and $j$ is calculated using the following expressions:

$$A_{ij} = \frac{M_{kj}}{\sum_{j=1}^{N} M_{kj}} \tag{1}$$

Adaptation is important for ecosystems to avoid coextinctions when confronted with natural disasters. Previous studies have assumed that link weight keeps unchanged, but this unchanged assumption has recently been questioned. Then, a structure-based approach has been proposed to compensate for the weight loss. Additionally, some studies suggested mechanisms to adjust link weights, and taking into consideration the dynamics of the systems. Recent studies emphasized the interactions importance between architecture and dynamic behaviors. Zhang et al. have put forward a co-adaptation model between structure and dynamics in mutualistic networks [37]. The UAV swarm is frequently run in harsh conditions and vulnerable to various disruptions, such as severe weather or opponents' defense. The objective of this research is to examine how a UAV swarm responds to disruptions and ensures its ability to withstand function loss or performance decline. Taking inspiration from the adaptive characteristics of ecosystems, we proposed four models: the static model, structural adaptation model, dynamic adaptation model, and co-adaptation model for the UAV swarm to withstand various disruptions. These four models are detailed presentation in Sections 2.1–2.4.

### 2.1. Static Model

In step 1 of static model, as depicted in Figure 2, we introduce a disruption to the swarm by removing one UAV (Number 7) marked with a red cross, along with all the links connected to this UAV ($M_{47}$, $M_{57}$), which are also marked with a red cross. In static model, removing a node does not result in any change in the interaction weight between the remaining nodes. And, the surveillance area of the remaining nodes remains stable.

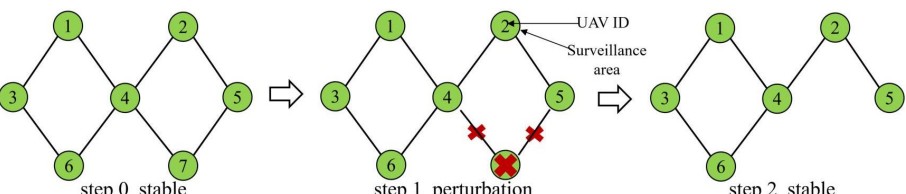

**Figure 2.** Static model.

### 2.2. Structural Adaptation Model

The first two steps of structural adaptation model are the same as the static model, as shown in Figure 3. After the attacked UAV is removed, in step 2, UAV 4 and 5 expand their surveillance area to account for the loss coverage of UAV 7. To simplify the process, when UAV $i$ is removed, the lost area is distributed evenly among its neighbors. Conse-

quently, for the remaining UAVs, the renewable surveillance area $y'_j$ is calculated using the following equations:

$$y'_j = \begin{cases} y_j \left( 1 + \dfrac{M_{ij}}{\sum_{k=1}^{N} M_{kj}} \right) & ,k \text{ is the neighbor of } j \\ y_j & ,otherwise \end{cases} \tag{2}$$

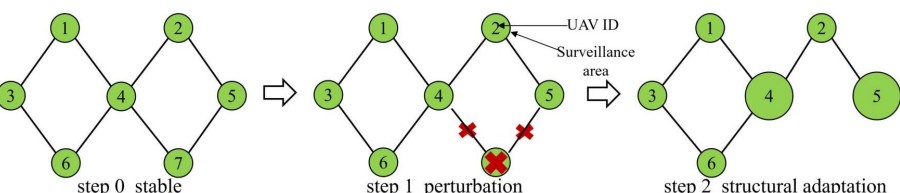

**Figure 3.** Structural adaptation model.

We call this model the structural adaptation model, as it solely depends on the network's structural information for adjusting the surveillance area. The model is not concerned with the dynamic nature of influence propagation. However, in real-world scenarios, the information exchange among the remaining UAVs also plays a crucial role in the surveillance mission, particularly during unexpected perturbations.

### 2.3. Dynamic Adaptation Model

Steps 0 and 1 of the dynamic adaptation model are still the same as static model. After removing UAV 7, UAVs 4 and 5, which are the neighbors of UAV 7, will first expand their surveillance areas to compensate for the loss surveillance areas. Then, UAVs 1, 2 and 6, the neighbors of UAVs 4 and 5 will narrow their surveillance areas to balance the overall surveillance mission. Similarly, the surveillance areas of the remaining UAVs will change based on their neighbors, showing a dynamic propagation of influence, as depicted in Figure 4. We refer to this mechanism as the dynamic adaptation model because the surveillance area is influenced by the dynamic propagation. We calculate the renewable surveillance area of $j$ using Equation (3).

$$y'_j = y_j + \sum_{i=1}^{N} \frac{y'_i - y_i}{N_{neighbor_i}} \tag{3}$$

$y_j$ is the initial surveillance area of $j$ before disturbance occurs. $i$ is one of the neighbors of $j$, which was influenced by disturbance earlier than $j$; $y'_i$ is the surveillance area of $i$ after disturbance influence take effect on $i$; $N_{neighbor_i}$ is the number of $i$'s neighbors, which are affected by the disturbance after $i$.

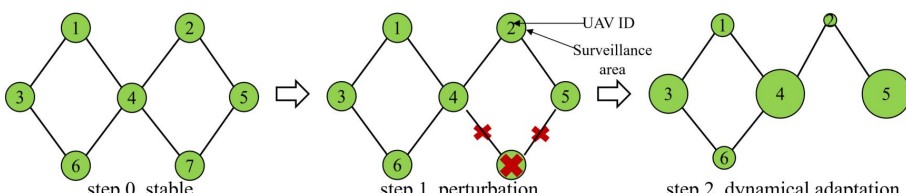

**Figure 4.** Dynamic adaptation model.

### 2.4. Co-Adaptation Model

In the co-adaptation model, we first repeat steps 0–2 of the structural adaptation model. Then, we proceed to apply step 2 of the dynamic adaptation model. Finally, the remaining UAVs have an equal surveillance area. We call this improved model as the co-adaptation model because we prioritize the implementation of the structural adaptation model and

then utilize dynamic adaptation, as shown in Figure 5. To calculate the renewable surveillance area ($y_j$) of UAV $j$ with $n$ UAVs removed, we employ the following equation.

$$y'_j = y_j \frac{N}{N - n} \tag{4}$$

$n$ denotes the quantity of removed UAVs; $N$ denotes the total UAVs number.

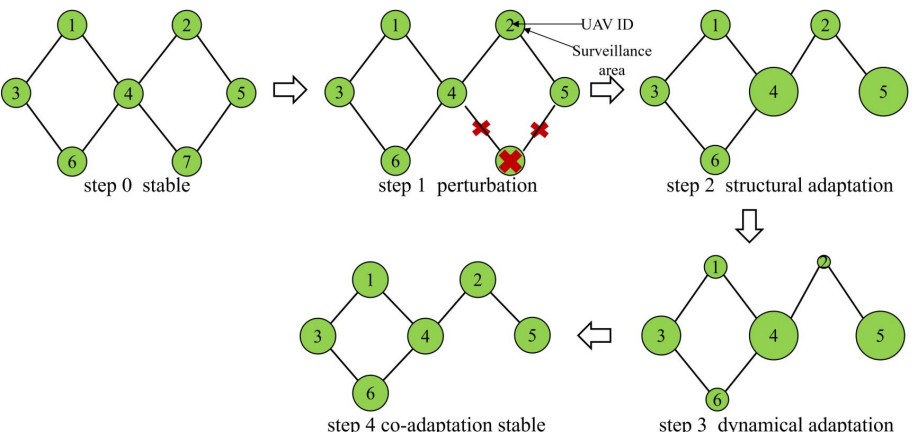

**Figure 5.** Co-adaptation model.

The results of static model and co-adaptation model have a similar characteristics, that the surveillance area of each remaining UAV is equivalent in each model. However, the surveillance area of each remaining UAV between two model are different. The surveillance area of each remaining UAV in static model method does not change during the mission, which means a overall performance declining after perturbation occurrence. In other words, under the static model, the swarm is not resilient. Conversely, the surveillance area of each remaining UAV in co-adaptation model changes according to the structure and the dynamic of neighbor UAVs, the overall performance of swarm maintains stability. That is, under the proposed model, the resilient swarm can be achieved. Although the last equivalent surveillance area characteristic is same, the response of two models is different. The static model is passive and un-resilient, co-adaptation model is active and resilient, which is helpful for withstanding disruptions and performance recovery.

## 3. Performance and Resilience Metric of Mission-Oriented UAV Swarm

In this section, we propose an improved resilience metric for mission-oriented UAV swarms, based on the metrics in [4,33]. In Section 3.1, we introduce the surveillance mission and choose the surveillance area as the system performance. Section 3.2 analyzes the resilience metric provided in [4,33] along with its limitations. In Section 3.3, we propose an improved resilience metric.

### 3.1. System Performance of Surveillance Missions

Consider the scenario of monitoring an unidentified battlefield zone using a UAV swarm [34], $UAV = [v_1, v_2, \ldots, v_n]$, (as depicted in Figure 6). $v_i$ performs surveillance mission independently, $r_i$ represents the radius of the surveillance area [31,32]. The mission performance reflects how effectively the swarm fulfills its mission [16]. In this study, the surveillance area of the targeted battlefield serves as the performance metric for each UAV. The system performance is the total surveillance area of the UAV swarm.

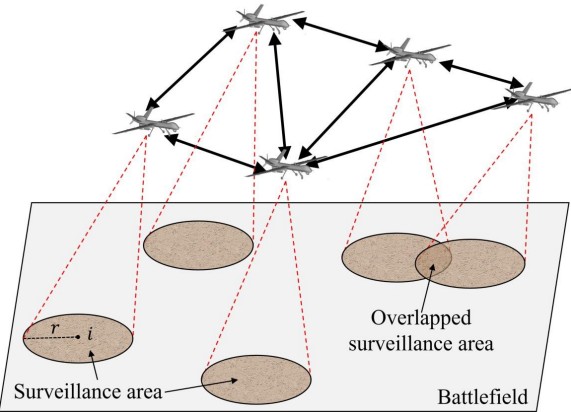

**Figure 6.** A collaborative surveillance mission utilizing a UAV swarm.

Each UAV diligently oversees the target area without interruption, constantly exchanging information with neighboring UAVs through communication links. The surveillance area for $v_i$ at a given time $t$ can be observed in Figure 7.

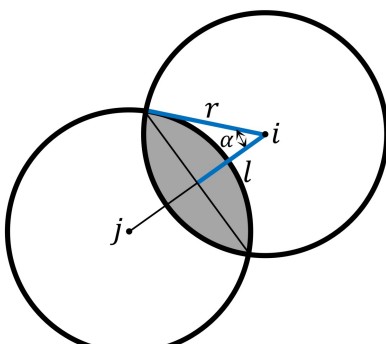

**Figure 7.** The overlap surveillance area of two adjacent UAVs.

In Figure 7, $r$ represents the radius of the surveillance target area for each UAV; $l$ represents the half distance between two adjacent surveillance areas; $i$ and $j$ are the centers of two surveillance areas; $\alpha$ denotes the included angle between $r$ and $l$. The shade area scanned by $i$ and $j$ occurs simultaneously.

In this paper, the system performance metric refers to the total surveillance area covered by a swarm of UAVs in a battlefield at time $t$. The total area covered by the UAV swarm during the surveillance mission is calculated by:

$$y(t) = \sum_{i=1}^{N_t} \left( \pi r_i^2 - \sum_{j=1}^{N_t} Su_{ij} \right) \tag{5}$$

$\pi r^2$ denotes the initial surveillance target area of each UAV before disruptions; $N_t$ denotes the total UAVs quantity at time $t$; $Su_{ij}$ denotes half of the shaded area in Figure 7, and is obtained through the following calculation:

$$Su_{ij} = \frac{\alpha_{ij}}{180} \pi r_i^2 - l_{ij} r \cdot sin\alpha_{ij} \tag{6}$$

$$l_{ij} = \frac{\sqrt{(d_{ix} - d_{jx})^2 + (d_{ix} - d_{jx})^2}}{2} \tag{7}$$

$$\alpha_{ij} = arccos\frac{l_{ij}}{r} \tag{8}$$

$d_{ix}, d_{jx}, d_{iy}, d_{jy}$ represent the position of $i$ and $j$. In this study, we compute the raw $y(t)$ by averaging multiple simulations. We subsequently employ the smooth algorithm [38] to derive the smooth performance values of $y(t)$.

### 3.2. Reported Resilience Metric

In [4,9,33], the swarm performance is determined by the total received information number at time $t$, which can be calculated by:

$$y(t) = \sum_{i=1}^{N_t} \sum_{j=1}^{R_i(t)} \Delta d_j^i \quad (9)$$

$d_j^i$ is the shortest path from $j$ to $i$; $\Delta$ is associated with the specific mission's requirement, with a value ranging from 0 to 1; $R_i(t)$ refers to the received information number of $i$ at time $t$, and $N_t$ denotes the total number of UAVs at time $t$.

In Figure 8, we present the performance value $y_t$ of a single UAV during a threat event from $t_0$ to $t_{final}$; the performance before the disruption is denoted as $y_D$, the minimum performance as $y_{min}$, and the performance after recovery as $y_R$; the time at the disruption occurs is $t_{th}$, the time at the system performance reaches its minimum as $t_{min}$, and the time at system recovery begins as $t_{ss}$; the values of $y_D$, $y_{min}$ and $y_R$ are calculated by

$$y_D = \frac{\sum_{t_0}^{t_{th}} y(t)}{t_0 - t_{th}}, y_{min} = \frac{\sum_{t_{min}}^{t_{ss}} y(t)}{t_{ss} - t_{min}}, y_R = \frac{\sum_{t_{ss}}^{t_{final}} y(t)}{t_{final} - t_{ss}}.$$

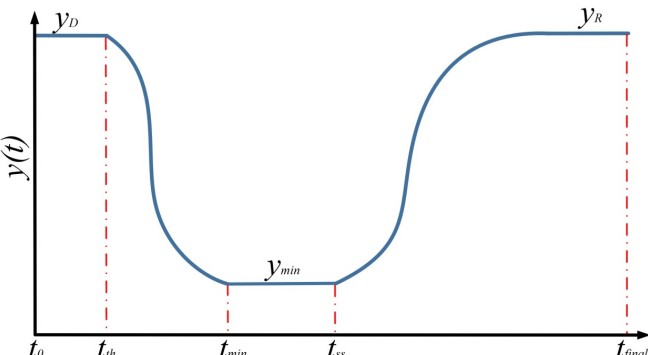

**Figure 8.** Schematic of single UAV performance curve.

Based on $y_D$, $y_{min}$, $y_R$, $t_{th}$, $t_{min}$ and $t_{ss}$, Tran et al. introduce performance factor ($\alpha$), absorption factor ($\delta$), recovery factor ($\rho$), recovery time factor ($\tau$), volatility factor ($\varsigma$), and give the calculation formulas as: $\sigma = \frac{\sum_{t_0}^{t_{final}} y(t)}{y_D(t_{final} - t_0)}$, $\delta = \frac{y_{min}}{y_D}$, $\tau = \frac{t_{ss} - t_0}{t_{final} - t_0}$, $\rho = \frac{y_R}{y_D}$, $\varsigma = \frac{1}{1 + exp[1 - 0.25(SNR_{dB} - 15)]}$.

Furthermore, Tran et al. propose a resilience metric for each disruption-recovery event.

$$S_i = \begin{cases} \delta\rho[\delta + \varsigma + 1 - \tau^{\rho - \delta}] & , \delta < \rho \\ \delta\rho[\delta + \varsigma] & , otherwise \end{cases} \quad (10)$$

Finally, the resilience value of a system is calculated by [3]:

$$S_{total} = \sum_{i=1}^{N_{threat}} \frac{w_{S_i}}{\sum_{j=1}^{N_t} w_{S_i}} S_i \quad (11)$$

$w_{S_i} = (1 - \beta)^{N_t - i}$ represents the weight of the $i_{th}$ event; $\beta$ represents the weight factor of $w_{S_i}$; $N_{threat}$ represents the number of disruption-recovery events; $N_t$ represents the total UAVs number at time $t$.

The assumption that each UAV is capable of autonomously selecting the shortest path to their targeted node in the swarm is impractical. Furthermore, a significant portion of the

information transmitted within a swarm is often redundant for conducting surveillance missions. Using the shortest path to calculate the total amount of information in a UAV swarm network as the system performance may not be suitable. Additionally, the metric of total information received fails to provide insight into the occurrence of surveillance area overlap, which is common in surveillance missions.

### 3.3. The Improved Resilience Metric

The goal of a resilience plan for a UAV swarm is to achieve the assigned task while enhancing anti-interference capabilities and improving performance even after disruption. An optimal resilience metric should encompass the entire process, from pre-damage preparation to post-recovery evaluation. Based on the resilience metric proposed in [4], we present an improved strategy for measuring resilience.

We use the MATLAB to obtain performance data through Monte Carlo simulation. The minimum performance is the most crucial factor in mission execution. However, the decline stage is not taken into account in previous studies when discussing $y_{min}$ [4,9,33], which is an oversight. In this paper, we obtain the minimum performance ($y_{min}$) using the following expressions

$$\overline{y_{min}} = \frac{\sum_{t_{th}}^{t_{ss}} y(t)}{t_{ss} - t_{th}} \tag{12}$$

To assess the performance and resilience curve of each disruption-recovery event accurately, it is more suitable to consider the time of disruption occurring rather than $t_0$, when discussing the recovery time factor. This is due to the fact that the timeframe $[t_0, t_{th}]$ has little impact on the performance of the disruption-recovery event.

$$\overline{\tau} = \frac{t_{ss} - t_{th}}{t_{final} - t_{th}} \tag{13}$$

Based on the $\overline{y_{min}}$ and $y_D$, the absorption factor can be obtained by:

$$\overline{\delta} = \frac{\overline{y_{min}}}{y_D} \tag{14}$$

Based on the renewal factors, the calculating formula for resilient value of each disruption-recovery event is as follows:

$$\overline{S_i} = \begin{cases} \delta\rho\left[\overline{\delta} + \varsigma + 1 - (\overline{\tau})^{\rho-\overline{\delta}}\right] & , \overline{\delta} < \rho \\ \delta\rho\left[\overline{\delta} + \varsigma\right] & , otherwise \end{cases} \tag{15}$$

## 4. Results

In this section, we conduct some experiments involving a UAV swarm performing a surveillance task in a battlefield. In Section 4.1, we provide a brief overview of the simulation experiment settings. Then, we give the experiments results in Section 4.2 and compare them with existing research to identify differences. Additionally, we discuss the factors that may have contributed to these discrepancies.

### 4.1. Experiment Description

Our proposed method is designed for a scenario in which a UAV swarm conducts a surveillance task over an unknown battlefield area (with a size of S = 1000 × 1000). UAVs within the swarm are identical, physical characteristics like weight, size, and shape are disregarded. During the mission, each UAV collaborates with other UAVs through multi-hop wireless communication, continuously sharing information to successfully accomplish the surveillance mission. It also considers the possibility of the UAVs being attacked by the enemy with a probability [39]. To establish the initial topology of the swarm system, a scale-free network with a preferential attachment algorithm is utilized [9,33], with $m_0 = 2$

and *m* = 2. The UAVs are initially scattered around the battlefield at the mission starting, and subsequently move as a random walk pattern during the mission (as shown in Figure 9).

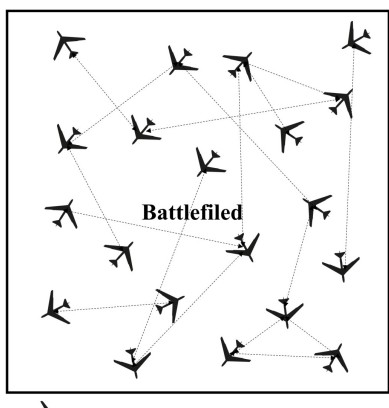

**Figure 9.** Surveillance mission over a specified battlefield.

The mission background is set and simulated using MATLAB R2022a. To mitigate the stochastic influence of the simulation, each case is run for 10 repetitions. The stochastic in the experiments is generated through the scale-free algorithm, message generation, and UAVs removal. The surveillance area of the swarm is calculated using Equation (5). The parameter settings for the surveillance mission are: $N = 20$, $r = 5$, $v = 10$.

*4.2. Experiment Results*

Firstly, we study the impact of communication links number (network density) on system performance, measured by the total number of information received in swarm [33]. In this case, four UAVs and their links are removed at the 100th simulation step as a disruption, followed by a recovery action at the 200th simulation step where two randomly selected UAVs and their links are restored. Figure 10 shows the simulation results; it reveals that the performance of 19 links exhibits a worse behavior, indicating a inferior ability to resist disruptive events and uncertain threats. We repeated the experiments with 2, 6, 8, 10, 12, 14, 16, and 18 UAVs removing. The results were consistent with Figure 10. These findings contradict the fact that the network density has little influence on surveillance missions. Therefore, the performance metric of messages total number proposed by Tran is inapplicable for UAV swarms when conducting surveillance missions.

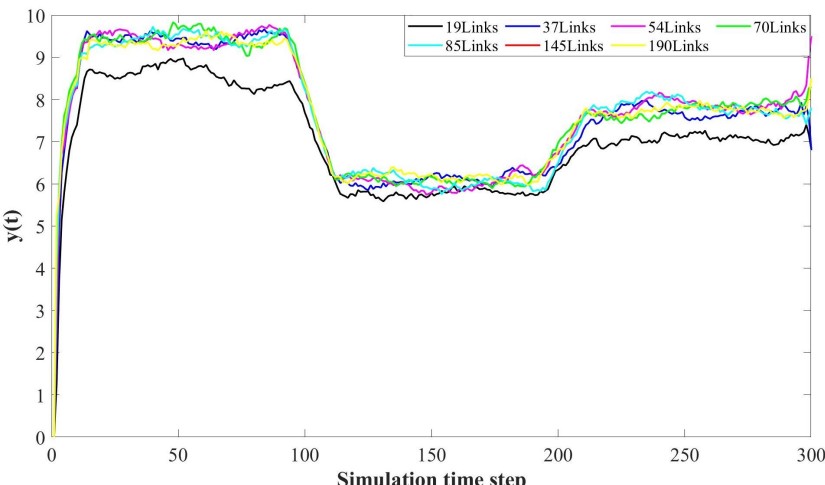

**Figure 10.** $y(t)$ as total messages under different communication links numbers.

Then, we calculate the total surveillance area by Equation (5). Typically, administrators expect $y(t)$ to remain steady during mission, indicating that the surveillance capacity keeps stable and the swarm is resilient. The simulation results for UAV numbers $N = 20$, $N = 18$, $N = 16$, $N = 14$, and $N = 12$ are displayed in Figure 11. In Figure 11, $y(t)$ maintains stability during the mission. Evidently, the surveillance area $y(t)$ decreases as the UAVs number declines, implying a decline in the surveillance performance of the UAV swarm. The $N = 20$ case denotes undamaged UAVs, and $y(t)$ hovers around the theoretical value 1570, aligning with the actual scenario. Thus, the simulation results depicted in Figure 11 validate our proposed model and performance metric are suitable for surveillance tasks compared with reference [3,4,9,33,39]. In Figure 11, the $N = 18$ case is relatively closer to the undisturbed event ($N = 20$). This is due to the fact that both 20 and 18 UAVs are sufficient for the surveillance mission. As for the remaining three cases ($N = 16$, $N = 14$, $N = 12$), $y(t)$ remains lower. This can be explained by Equation (5) where it is evident that $y(t)$ decreases with decreasing values of $N_t$.

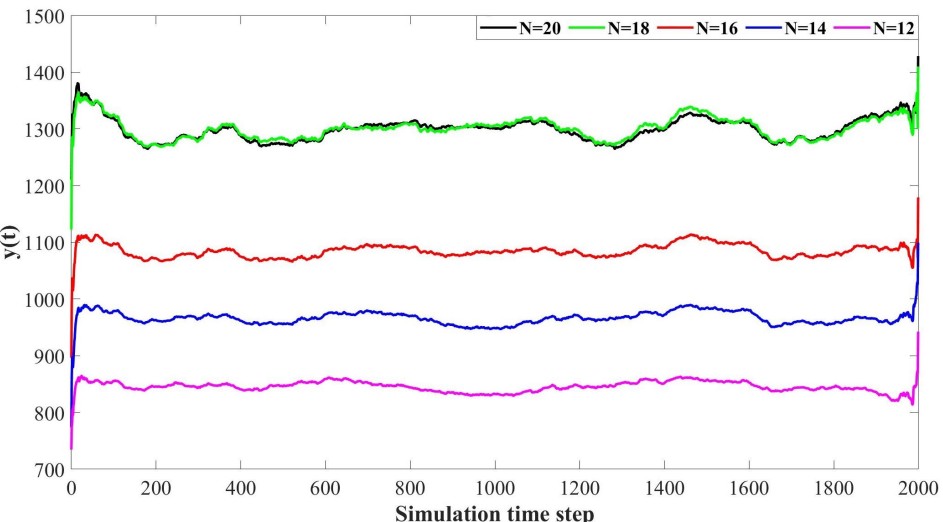

**Figure 11.** $y(t)$ as surveillance area under different disturbance programs.

Furthermore, we compare the static model, structural adaptation model, dynamic adaptation model and co-adaptation model in the context of varying numbers of UAVs (2/4/6/8/10/12/14/16/18) and related links being removed at 200/400/600/800/100/ 1200/1400/1600/1800 simulation steps, respectively. For comparison, we also depict the non-disturbance results in Figure 12. In non-disturbance case, the performance remains stable at around 1400. Contrarily, the performance of static case exhibits a continuous decline trend. In other words, under the static model, the swarm is not resilient. The structural adaptation model and dynamic adaptation model can enhance the performance in a limited manner compared with static model. As shown in Figure 12, the surveillance performance of the UAV swarm with the co-adaptation method keeps a high level and closes to non-disturbance case, which shows a satisfactory resilience character. When the number of removed UAVs exceed 10, the performance experiences a downward slide. However, the performance is still better than the three other models. This highlights the advantage of the co-adaptation model in enabling the swarm to withstand certain disruptive events and uncertain threats. In other words, under the co-adaptation model, a resilient swarm can be achieved.

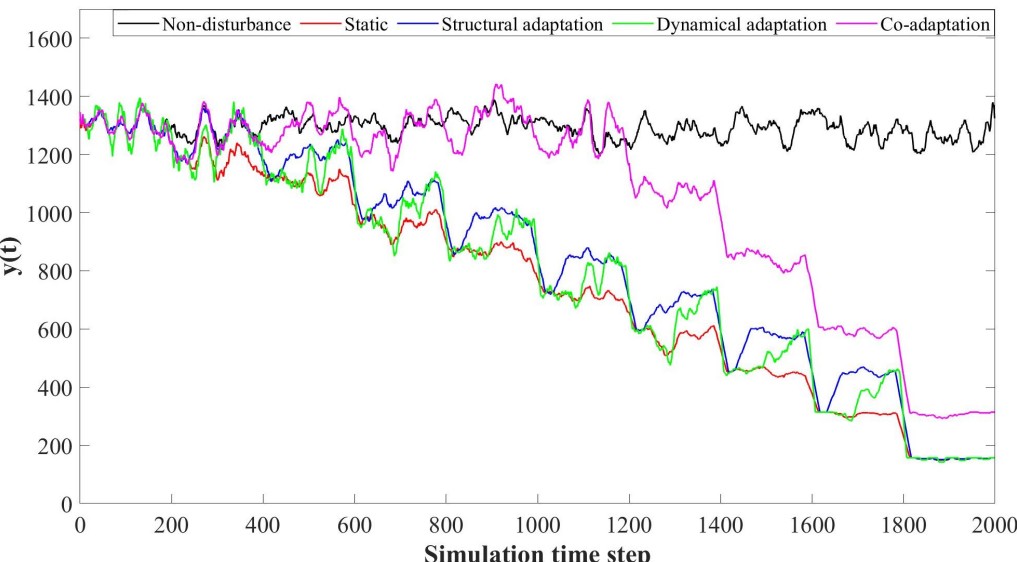

**Figure 12.** $y(t)$ as surveillance area under different models.

Secondly, we use Equation (10) to calculate the resilience value with Tran's method, and our proposed method for each individual disruption-recovery event across various quantities of communication links. The comparison results are shown in Figure 13. As illustrated in Figure 13a, the resilience values obtain by Tran's method display a subtle variance with the number of UAV communication links, but exhibit a noticeable decrease as the number of disruption-recovery events increases. Thus, Tran's resilience metric shows the tendency that the information exchange within the UAV swarm remains unaffected by network density. Actually, with a rise in UAV communication links, the $d_j^i$ calculated by a shortest path algorithm (such as Dijkstra's algorithm) should be shorter. It is worth noting that an increased quantity of communication links only contributes to performance and not to the resilience value, as demonstrated in Figures 10 and 13a. Hence, it can be concluded that the approach proposed by Tran is not suitable for evaluating surveillance missions.

Contrarily, the resilience metrics in Figure 13b are obtained by our proposed method, which show a notable consistency in the resilience values across different numbers of communication links. This suggests that the network density has no impact on the resilience value in a series of disruption-recovery events. Additionally, the resilience value decreases in line with the removal of UAVs and communication links, which aligns with the surveillance performance trend illustrated in in Figure 11. So, our proposed resilience metric is suitable for evaluating surveillance mission.

Finally, the total resilience value of the UAV swarm measured by Tran's method and our proposed method is calculated using Equation (11) based on the resilience value obtained by Equations (10) and (15). The results are shown in Figure 14. As depicted in Figure 14, there is a decreasing trend in the total resilience values calculated using Tran's method. This trend contradicts the fact that the number of communication links in a UAV swarm has little influence on the system's surveillance performance. In contrast, the total resilience values calculated by our proposed method for different network densities scatter around a fixed value, and do not exhibit such a trend. This validates the feasibility of our proposed model in enabling a UAV swarm to efficiently carry out the assigned surveillance mission in a targeted battlefield zone. So, compared with the method and metric proposed in [3,4,9,33,39], our model and metrics are more suitable for a UAV swarm during conducting a surveillance mission.

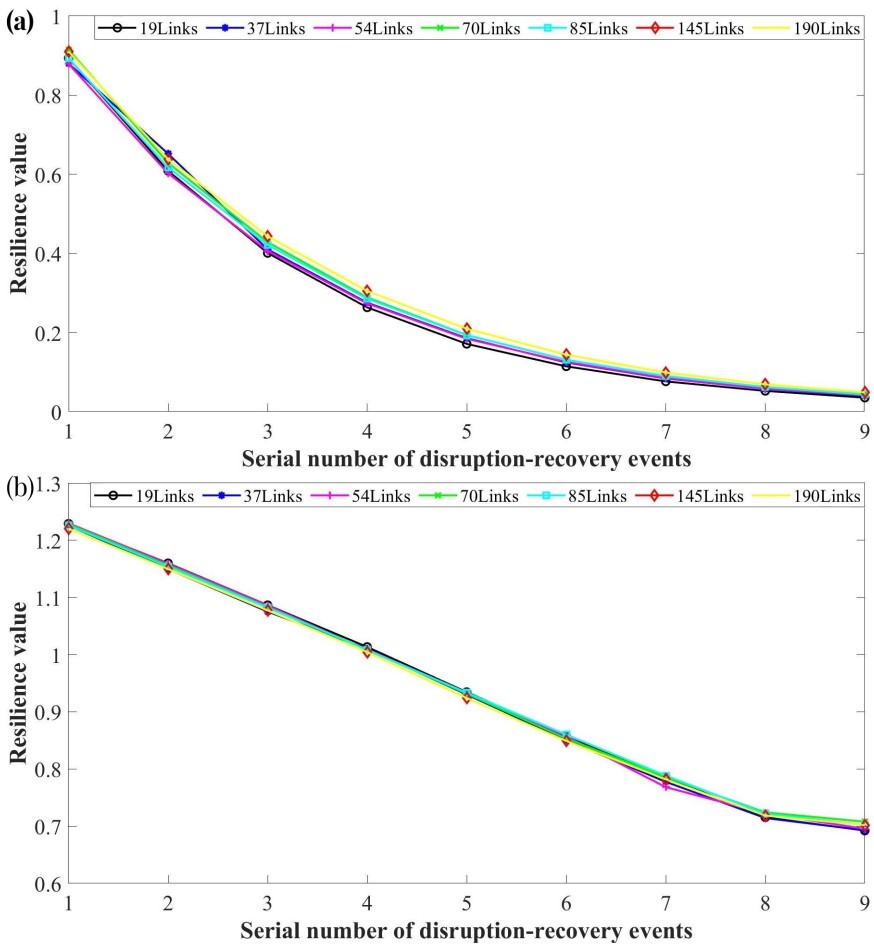

**Figure 13.** Resilience value obtained via Tran et al. [4] (**a**) and our for proposed method (**b**) for each event.

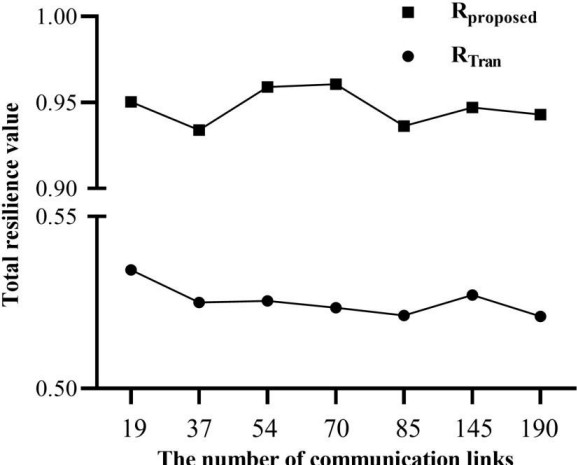

**Figure 14.** Contrast between total resilience value of our proposed method and Tran.

## 5. Conclusions

UAVs are expected to quickly respond to dynamic changes that may occur during missions. Major causes of mission failures are inevitable disruptive events and uncertain threats. To address this, we have developed an improved model for UAV swarms that considers the impact of network structure and dynamic adaptation within the swarm. This improved model enables the swarm to better cope with adverse events and restore its performance. Based on the co-adaptation model, we also proposed a renewed resilience

metric. This metric measures the disparity between the minimum performance observed after a disruption event and the standard performance before the disruption. It provides insights into the swarm's ability to withstand disruptions, recover its desired performance, and effectively accomplish its assigned task.

This model is particularly useful for simulating and evaluating the resilience of mission-based UAV swarms. It focuses on system designs that include surveillance performance and the ability to withstand uncertain threats, as well as adaptability to potential recovery actions. The resilience of the system, measured by $S_i$, provides a general understanding of its performance by assessing its ability to maintain and restore surveillance capabilities. By analyzing $S_{total}$, which considers stochasticity in simulated performance and system adaptation over time, direct comparisons of system resilience can be made. The results of this study indicate that co-adaptation improves both system performance and resilience. However, it is important to note that the resilience gained by co-adaptation is reduced when the surveillance area exceeds the capacity of the UAV swarm. These findings can be valuable in guiding future investment decisions and research agendas in the overall design process. For instance, when combined with capacity analysis, UAV swarm designers can use these assessments to determine if co-adaptation in the surveillance area is a worthwhile task for their system, based on knowledge of specific threat likelihoods.

In this study, we did not consider certain limitations, such as the surveillance capacity of each UAV and the increasing probability of attacks with the surveillance area expands. Furthermore, we recognize that communication distance and time delay can also impact the co-adaptation capacity of our resilience enhancement model for UAV swarms. In future research, we intend to study the influence of surveillance capacity, communication distance, and time delay on our proposed model.

**Author Contributions:** Conceptualization, T.Z. and K.W.; methodology, K.W.; software, K.W.; validation, K.W.; formal analysis, T.Z.; investigation, T.Z.; resources, T.Z.; data curation, C.Z.; writing—original draft preparation, K.W.; writing—review and editing, K.W.; visualization, K.W.; supervision, T.Z.; project administration, C.Z. All authors have read and agreed to the published version of the manuscript.

**Funding:** This research received no external funding and The APC was funded by T.Z.

**Data Availability Statement:** The data presented in this study are available on request from the corresponding author.

**Acknowledgments:** We extend our gratitude to four reviewers who provided comments to our manuscript.

**Conflicts of Interest:** The authors declare no conflicts of interest.

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
