# Peer review of "A Co-Adaptation Method for Resilience Rebound in Unmanned Aerial Vehicle Swarms in Surveillance Missions"

_drones, doi:10.3390/drones8010004_

Round 1
Reviewer 1 Report
Comments and Suggestions for Authors Thank you very much! I would suggest that the English writing of this paper must be improved. Please have a proofread at your university English writing center or using a professional English service before the future revision submission. There are too many grammar errors in the current paper and several technical words are not used properly. In general, for reviewers, it will be hard for us to review a paper with so many grammar errors and still keep reviewing :-) In addition to the English writing, the contents of the paper also need to be improved, especially getting all the equations proven, and having the assumptions more reasonable. In summary, this paper is not ready for publishing yet. 1) A brief summary: This paper provides a new method to maintain the resilience of a UAV swarm and the metrics for the system performance of a UAV swarm in the surveillance mission.2) General concept comments: The proposed methods and metrics are reasonable, but the formula (i.e., equations) listed in the paper may not be proven, which must be improved in the future revisions.
3) Specific comments
a) Line 215: Please explain why the surveillance coverage area of a UAV is in a circle? it could be in a rectangle too.
b) Line 273: Please prove the equations 12, 13, 14, 15.
c) Line 379: Please provide how resilience is determined. The equations 10 and 15 may not be good enough to calculate the total resilience value of the swarm system in the surveillance mission. Best wishes and have a nice day, Comments on the Quality of English Language
In addition, the English writing of the paper should be improved. Please have a proofread at your university English writing center or using a professional English service.
Author Response
Dear reviewers,
Thank you very much for your kindly comments on our manuscript. There is no doubt that these comments are valuable and very helpful for revising and improving our manuscript.
Name: Kunlun Wei
E-mail: weiklun@mail2.sysu.edu.cn

Reviewer 2 Report
Comments and Suggestions for Authors
The paper aims to improve the resilience of UAV swarms during missions by developing a co-adaptation model that combines structural and dynamic aspects of the swarm. The proposed model helps the swarm recover from unexpected disruptions and threats, ultimately enhancing its ability to withstand and adapt to unforeseen challenges. The paper also introduces a resilience metric to measure the swarm's performance recovery. Through simulation experiments, it demonstrates how this approach can help UAV swarms bounce back to their initial performance levels after disruptions, making them more robust in the face of unpredictable threats and disruptions.
Overall, the paper presents a promising approach to enhancing the resilience of UAV swarms, but it would benefit from improvements in presentation, explanation of results, and a more in-depth discussion of its relevance in the existing research landscape. In my opinion, the paper has some notable strengths but also some areas where improvement is needed:
1. The paper's Introduction should provide a more clear and concise presentation of the problem being addressed and the contributions of the study. It would be beneficial to explicitly state what specific issues the proposed co-adaptation model and resilience metric aim to solve. It is expected that the contribution and novelty presented in this paper will be clearly explained in the Introduction section. A general literature is presented, but what problem is being solved? What is the suggestion? It is expected that the authors will objectively explain and present the contributions suggested by this study at the end of the induction.
2. The Materials and Methods section needs to be structured in a more reader-friendly manner. It should start with a general overview and, if possible, include a flow diagram to provide a visual roadmap of the method. Additionally, a graphical abstract for the method could enhance comprehension.
3. Figure 1 could benefit from more detailed annotations and a legend to make it more accessible to readers. Consider breaking it down into sub-figures for a more detailed presentation.
4. There may be a numbering error in the headings. It seems that heading number 3 might be related to the method and should be correctly numbered and positioned.
5. Sharing videos or animations of the simulations as evidence would be a valuable addition to the paper, providing a more concrete understanding of the results.
6. In Figure 8, the periodic effect is not explained. The paper should address the reason for this effect and consider whether seasonal effect elimination was necessary for a more accurate analysis.
7. The placement of graphics in the Conclusion section should be closely related to the content being explained to enhance clarity.
8. The Discussion section of the paper needs more depth, particularly in terms of comparing the results and findings with similar or related studies. This would provide a broader context for the research and help readers better understand its significance.
Author Response

(The authors gave the same response as above.)

Reviewer 3 Report
Comments and Suggestions for Authors
1) Provide a list of contributions and give sufficient evidences to support your claim.
2) Figure (1): static model and co-adaptation model lead to the same result; however, static model does that in only two steps. I believe the authors need to explain this and/or probably use another example to show the advantages of the co-adaptation model.
3) Equation (12): what happens if t_ss=t_th?
4) Figure (8): It seems that the co-adaptation model is diverging! Is that correct? The authors need to increase simulation time to see the steam-state behavior.
Comments on the Quality of English LanguageAcceptable.
Author Response

(The authors gave the same response as above.)

Reviewer 4 Report
Comments and Suggestions for Authors
This paper aims to propose a co-adaptation model that combines the structure with the dynamic characteristics of the UAV swarm to quickly respond to dynamic changes that may occur during mission execution. The authors then propose a renewed resilience metric, built upon the developed model, to measure factors that were not previously included, such as the disparity between the minimum performance observed after a disruption event and the standard performance before the disruption. This furnishes insights into the swarm’s ability to withstand disruptions.
As the study aims to explore how the UAV swarm responds to disruptions, the manuscript offers a detailed description of four distinct models designed to address these disruptions. The objective is to demonstrate the superiority of the co-adaptation model. Simulation results are presented to showcase the feasibility and effectiveness of the proposed approach, thereby validating the modified metric.
Overall, the paper provides new and interesting insights into effectively addressing the identified issues. The conceptualization of the model is robust, and the presentation, along with technical descriptions, is clear and satisfactory.
The paper is well-written and engaging to read, offering sufficient details and presenting the results appropriately. However, some minor issues should be resolved before publishing:
- It would be useful to indicate whether the four models described in Section 2.1–2.4 are defined by the authors or if they are retrieved from gold standards. If the authors defined these models, a discussion is needed on the sequential formalization process, addressing how and why these models were developed and the specific problems they aim to overcome.
- Figure 8 might benefit from a comparison between all defined models, not just the static and co-adaptation models. Otherwise, the definitions in Section 2 are mostly useless.
- In the Conclusions section, briefly discuss how the proposed model could be applied practically by including suggestions and a strategy to incorporate the model. Remember that the goal of the "Conclusions" section is not just to summarize the work but to emphasize its significance and potential impact.
- Finally, please carefully proofread and spell-check to eliminate typos. A minor revision is needed to improve the quality of English. Here are some specific suggestions:
1. Use a semicolon (;) instead of a period (.) to separate different parameter definitions in mathematical formulas (lines 183-185, 219-221, etc.).
2. Delete the standard sentence in Author Contributions: “For research articles with several authors, a short paragraph specifying their individual contributions must be provided. The following statements should be used.”
3. Clarify the meaning of the term “coincides” in the Fig. 3 caption: “The surveillance area of two adjacent UAVs coincides.” What exactly does "coincides" mean in this context?
To sum up, the paper is promising but requires some revision and improvement.
Comments on the Quality of English LanguageCarefully proofread and spell-check to eliminate typos. A minor revision is needed to improve the quality of English.
Here are some specific suggestions:
1. Use a semicolon (;) instead of a period (.) to separate different parameter definitions in mathematical formulas (lines 183-185, 219-221, etc.).
2. Delete the standard sentence in Author Contributions: “For research articles with several authors, a short paragraph specifying their individual contributions must be provided. The following statements should be used.”
3. Clarify the meaning of the term “coincides” in the Fig. 3 caption: “The surveillance area of two adjacent UAVs coincides.” What exactly does "coincides" mean in this context?
Author Response
Dear reviewers,
Thank you very much for your kindly comments on our manuscript. There is no doubt that these comments are valuable and very helpful for revising and improving our manuscript. Following your comments, we have revised and improved the manuscript accordingly, and all changes have been indicated in yellow. We hope that the revised manuscript is satisfactory.
Name: Kunlun Wei
E-mail: weiklun@mail2.sysu.edu.cn

Round 2
Reviewer 1 Report
Comments and Suggestions for Authors
This paper revision has been greatly improved. With a few of minor modifications, this paper can be published. Please work on the following specific comments:
Line 57: please define "SDN"
Line 79: please use "could not".
Line 85: please keep consistent with the verb tense in the paragraph.
Line 121: please use past tense or present tense for all these items.
Line 206: please use "does not".
Thanks!

Please have another proofread to work on the verb tense as mentioned in the commented paper V2 in the attachment.
Author Response
Dear Reviewer:
We are very grateful to the Reviewers for your time and careful assessment of our manuscript. The comments and suggestions provided are very helpful for us to improve the quality of our manuscript. We have responded to all the comments as follows, and revised our manuscript accordingly. We hope that the final manuscript is satisfactory. Attached are the detailed responses to the reviewers’ comments. Thank you again for your kind help and guidance.
Thanks you and best regards.
Yours Sincerely
Kunlun Wei
Email: weiklun@mail2.sysu.edu.cn

Reviewer 2 Report
Comments and Suggestions for Authors
Paper can be accepted in present form
Author Response
Dear Reviewer:
Thank you very much for your positive comments, and your recognition to our improvements is much appreciated. We are very grateful for your time and efforts throughout the review process. Your proposed constructive comments and suggestions help us improve the quality of this manuscript, and also open up some ideas for our future work.
Thanks you and best regards.
Yours Sincerely
Kunlun Wei
Email: weiklun@mail2.sysu.edu.cn
Reviewer 3 Report
Comments and Suggestions for Authors
No further comment.
Comments on the Quality of English LanguageAcceptable.
Author Response
Dear Reviewer:
Thank your very much for your time and efforts throughout the review process. Your proposed constructive comments and suggestions are useful for improving the quality of this manuscript. Thank you again for your kind help and guidance. We also are grateful that you are satisfied with our revision.
Thanks you and best regards.
Yours Sincerely
Kunlun Wei
Email: weiklun@mail2.sysu.edu.cn